# Toxicological Effects of Titanium Dioxide Nanoparticles on Human Menstrual Blood Mesenchymal Stem Cells

**DOI:** 10.3390/ijms262211168

**Published:** 2025-11-19

**Authors:** Alberto Parra-Barrera, Rebeca López-Marure, Ernesto Romero-López, Claudia Camelia Calzada-Mendoza, José Arellano-Galindo, Ricardo Rangel-Martínez, Gisela Gutiérrez-Iglesias

**Affiliations:** 1Laboratorio de Medicina Regenerativa y Estudios del Cáncer, Departamento de Investigación y Posgrado, Escuela Superior de Medicina, Instituto Politécnico Nacional, Ciudad de México 11340, Mexico; 2Immunotherapy, Regenerative Medicine, Laboratorio QuantumLab, Departamento de Investigación, Puerto Vallarta 48313, Mexico; 3Departamento de Fisiología, Instituto Nacional de Cardiología Ignacio Chávez, Ciudad de México 14080, Mexico; 4Laboratorio de Señalización Celular, Departamento de Investigación y Posgrado, Escuela Superior de Medicina, Instituto Politécnico Nacional, Ciudad de México 11340, Mexico; 5Departamento de Medicina, Centro Interdisciplinario de Ciencias de la Salud Unidad Milpa Alta, Instituto Politécnico Nacional, Ciudad de México 12000, Mexico; 6Laboratorio de Virología Clínica Experimental, Unidad de Investigación en Enfermedades Infecciosas, Hospital Infantil de México “Federico Gómez”, Ciudad de México 06720, Mexico; 7Corporativo ExomeLab–Biotecnologia Exosomal, Ciudad de México 09479, Mexico

**Keywords:** nanoparticles, titanium dioxide, endometrial mesenchymal stem cells, in vitro cultures

## Abstract

Human exposure to titanium dioxide nanoparticles (TiO_2_ NPs) is common. These NPs are used in cosmetics, paint, food, and other products. Their nanometric size (<100 nm) allows entry into the bloodstream, from which they can reach organs and cells throughout the body. Although TiO_2_ NPs have been reported to damage certain cell lines and organs and to alter cellular function, their impact on human menstrual blood mesenchymal stem cells (hMB-MSCs) is unknown. This study evaluated the effects of TiO_2_ NPs on viability, proliferation, morphology, membrane-marker expression, and reactive oxygen species (ROS) production in primary cultures of hMB-MSCs derived from menstrual blood. Cells were exposed to different concentrations of TiO_2_ NPs for 3, 7, and 14 days. TiO_2_ NPs decreased hMB-MSC viability and proliferation in a concentration- and time-dependent manner. Cellular viability was reduced by up to 6%, 11%, and 18% at 3, 7, and 14 days, respectively (statistically significant vs. control). Cellular proliferation decreased by 3%, 5%, and 33% at 15.63, 62.5, and 250 μg/mL TiO_2_, respectively. TiO_2_ NPs were internalized and observed in the cytoplasm, forming perinuclear aggregates. NP-exposed cells showed reduced membrane expression of CD73 (7.9% decrease) and CD90 (25.72% decrease) compared with control cells. Finally, TiO_2_ NPs at 15.63, 62.5, and 250 µg/mL reduced ROS generation by 56.79%, 62.79%, and 53.35%, respectively, after 4 h (statistically significant vs. control). In summary, exposure to high concentrations of TiO_2_ NPs leads to intracellular nanoparticle deposits and alters key functions of human menstrual blood mesenchymal stem cells, including immunomodulation, immune protection, molecular behavior, cell differentiation, and regenerative capacity.

## 1. Introduction

Titanium dioxide nanoparticles (TiO_2_ NPs, <100 nm) are produced at large scale worldwide for use in aerosols, suspensions, and emulsions. These NPs are incorporated into many products, including paints, plastics, cosmetics, pharmaceuticals, food additives, and medical implants [1]. Occupational exposure occurs primarily through inhalation and dermal contact in manufacturing and processing settings. Consumers can also be exposed via ingestion of products that contain TiO_2_ (for example, some candies and chewing gum), and individuals with dental implants may experience localized exposure [2]. Once TiO_2_ NPs enter the body, they can reach the bloodstream, cross the endothelial barrier, and accumulate in multiple organs, including the kidneys, liver, lungs, spleen, brain, and reproductive organs [3]. At the cellular level, NPs are enveloped by membrane invaginations and internalized into the cytoplasm, where they interact with organelles, alter cellular structure and function, and may cause intracellular lesions [4]. TiO_2_ NPs have been reported to produce cytotoxic effects in various cell lines. Examples include inhibited proliferation of osteoblasts (MC3T3) [5] and impaired adhesion, migration, proliferation, and differentiation of mesenchymal stem cells (MSCs) [6]. TiO_2_ NPs also provoke inflammatory responses in endothelial cells [7] and trigger dose-dependent autophagosome formation in human keratinocytes (HaCaT) [8]. In rat Sertoli cells, TiO_2_ exposure induces apoptosis, elevates reactive oxygen species (ROS), decreases antioxidant enzymes (superoxide dismutase, catalase, glutathione peroxidase), lowers mitochondrial membrane potential, and promotes cytochrome-C release into the cytosol [6]. Since their initial characterization in bone marrow, MSCs have been a major focus of research because of their regenerative and immunomodulatory properties [8,9]. These clonogenic stromal cells can be isolated from connective tissue, bone marrow, adipose tissue, placenta, peripheral blood, synovial fluid, periosteum, and menstrual blood [9,10,11,12,13]. In bone marrow, MSCs coexist with hematopoietic stem cells but do not differentiate into hematopoietic lineages; rather, they are restricted to mesenchymal lineages, such as osteoblasts, chondrocytes, and adipocytes, and can contribute to supportive stromal functions [14]. Human menstrual blood mesenchymal stem cells (hMB-MSCs) express pluripotency- and differentiation-related markers (e.g., Oct4, SSEA-4, and c-kit) and can differentiate into chondrocytes, adipocytes, and osteoblasts [14]. They also display reported transdifferentiation potential toward cardiomyogenic, neurogenic, keratogenic, and hepatogenic phenotypes [15]. hMB-MSCs possess self-renewal capacity and produce substantial amounts of proteoglycans and type II collagen, which supports their candidacy for cartilage regeneration therapies [16]. Although TiO_2_ NPs have been shown to promote osteogenic differentiation in adipose-derived MSCs [11] and bone marrow MSCs [17], the specific effects of TiO_2_ NPs on MSCs derived from human endometrium remain incompletely characterized. Therefore, this study aimed to evaluate the in vitro effects of TiO_2_ NPs on hMB-MSCs, with a focus on viability, proliferation, morphological changes, membrane-marker expression, and ROS production.

## 2. Results

### 2.1. Characterization of TiO_2_ NPs

The assessed TiO_2_ NPs were confirmed to be free of endotoxin and contained only titanium and oxygen. Our research group previously characterized these TiO_2_ NPs [18]. NPs were suspended in medium, either supplemented with 10% FBS or serum-free. When suspended in serum-supplemented culture medium, the primary particles presented as spherical structures smaller than 50 nm, but they formed aggregates with a size distribution between 105 and 1281 nm (mean size 421 nm). The ζ-potential was −6.98 mV, although this value varied depending on the suspension medium (see Appendix A). TiO_2_ NPs exhibited a BET surface area of 46.85 m^2^/g. The powdered TiO_2_ material showed a narrow primary-particle size distribution and relatively homogeneous morphology, with individual particle sizes ranging approximately from 9 to 25 nm; a predominant size of 19 nm was evident in the SEM images (Figure 1A,B). In PBS suspensions, SEM revealed clusters and aggregates larger than 100 nm (red arrows) alongside semi-round, dispersed individual particles (Figure 1C). In DMEM/F12 with 10% FBS, single particles measured between 9 and 14 nm (yellow arrows) (Figure 1D). X-ray diffraction analysis indicated the TiO_2_ NPs composition was 96% anatase and 4% rutile.

### 2.2. Characterization of Human Menstrual Blood Mesenchymal Stem Cells

The characterization of human menstrual blood-derived mesenchymal stem cells (hMB-MSCs) followed the standardized protocol of the International Society for Cell Therapy (ISCT). Cells grew as a monolayer and formed a homogeneous population with the characteristic spindle-shaped morphology. Cell size ranged from 10 to 20 µm, with a long, flattened, fibroblast-like appearance (Figure 2A). Differentiation potential was assessed after 14 days of induction. Cells differentiated into adipogenic and osteogenic lineages: adipocytes containing lipid droplets were visualized (Figure 2B), and osteoblasts showed extracellular calcium deposits (Figure 2C). Flow cytometry analysis was used to assess surface marker expression according to ISCT criteria. Hematopoietic markers were minimally expressed: CD14 (1.27%), CD34 (1.85%), CD45 (0.04%), and HLA-DR (1.98%). MSC markers showed high expression for CD73 (88.75%) and moderate expression for CD90 (67.19%), whereas CD105 expression was low (4.88%) (see Appendix A). Taken together, the population isolated from menstrual blood displays characteristic morphology and multilineage differentiation capacity consistent with hMB-MSCs; however, CD105 expression levels do not meet the typical ISCT thresholds and are noted below.

### 2.3. Viability and Cell Proliferation Assays

hMB-MSCs were initially treated with TiO_2_ NPs at 500 µg/mL and then serially diluted down to 0.2 µg/mL. Viability was assessed at 3, 7, and 14 days. Exposure to TiO_2_ NPs reduced hMB-MSC viability in a concentration- and time-dependent manner. At 500 µg/mL, viability decreased by 6% (day 3), 11% (day 7), and 18% (day 14); these reductions were statistically significant versus control and between treatment times. At 0.2 µg/mL, viability declined by 3% (day 3), 7% (day 7), and 7% (day 14), changes that were not statistically significant compared with the control (Figure 3A). Based on these results, three concentrations (15.63, 62.5, and 250 µg/mL) were selected for all subsequent experiments. For the proliferation assay, hMB-MSCs were treated with these concentrations for 7 days. Exposure to TiO_2_ NPs at 250 µg/mL for 7 days reduced cell proliferation by 33% versus control. At 15.63 and 62.5 µg/mL, proliferation was only slightly inhibited (3% and 5%, respectively) (Figure 3B). A correlation was observed between decreased proliferation and reduced viability induced by TiO_2_ NPs.

### 2.4. Internalization of TiO_2_ NPs by hMB-MSCs

hMB-MSCs were treated with TiO_2_ NPs at 15.63, 62.5, and 250 µg/mL for 7 days. After treatment, cells were stained with safranin to examine nanoparticle internalization. Light microscopy showed a centrally located nucleus with three to four nucleoli in hMB-MSCs (Figure 4A). TiO_2_ NPs were localized in the cytoplasm and formed aggregates adjacent to the nuclear membrane (Figure 4B,D), with aggregation markedly more pronounced at 250 µg/mL (Figure 4D). To assess ultrastructural effects, cells treated with the same concentrations were processed for transmission electron microscopy (TEM) and analyzed after 3 days. Nuclear membranes remained intact (Figure 4E), while aggregates of varying sizes were observed throughout the cytoplasm (Figure 4F) and within cytoplasmic vacuoles (blue arrows, Figure 4G). At 250 µg/mL, there was pronounced disintegration of plasma membranes and organelles (Figure 4H); similar but less severe damage was evident at 15.63 µg/mL. Aggregate sizes ranged approximately from 100 to 241 nm, and marked cytoplasmic vacuolization was observed at all tested concentrations (15.63, 62.5, and 250 µg/mL).

### 2.5. Expression of Membrane Markers on hMB-MSCs Treated with TiO_2_ NPs

hMB-MSCs were treated with 62.5 µg/mL TiO_2_ NPs for 24 h. Surface-marker expression was assessed by flow cytometry using FITC- or PE-conjugated antibodies against CD73, CD90, CD105, CD14, CD34, CD45, and HLA-DR. Compared with control cells, TiO_2_ exposure reduced membrane expression of CD73 by 25.72% and CD90 by 7.9% (see Appendix A).

### 2.6. TiO_2_ NPs Reduced ROS Production in hMB-MSCs

hMB-MSCs were exposed to 15.63, 62.5, and 250 µg/mL TiO_2_ NPs for 4 days. After exposure, cells were subjected to the nitroblue tetrazolium (NBT) assay to detect superoxide production via NADPH oxidase activity. Compared with controls, TiO_2_-treated hMB-MSCs showed a notable decrease in NBT reduction (Figure 5), indicating reduced NADPH-oxidase activity and lower superoxide/ROS production. This decrease in ROS signaling could imply altered innate-defense responses, which may affect cellular susceptibility to bacterial or viral challenges.

## 3. Discussion

The global population is widely exposed to titanium dioxide nanoparticles (TiO_2_ NPs) in one form or another. Cellular and molecular interactions with these particles can plausibly affect human health, which makes it essential to deepen our understanding of TiO_2_ toxicity mechanisms [19]. Given the increasing use of TiO_2_ NPs in consumer products and the evidence of nanoparticle-induced damage in multiple human cell lines, it is critical to determine whether and how human tissues are affected by such exposures [20]. For this reason, we examined the in vitro effects of TiO_2_ NPs on hMB-MSCs.

We used Alamar Blue to assess cell viability because it is a sensitive, rapid colorimetric method commonly used for nanotoxicity screening [21]. At the highest tested dose (500 µg/mL), TiO_2_ exposure reduced hMB-MSC viability by 6% at day 3 and by 18% at day 14. At 0.2 µg/mL, maximal mortality reached 7% by day 14 (Figure 2A). These results indicate a concentration-dependent effect on viability and are consistent with prior reports [22]. Occupational inhalation and oral exposure to TiO_2_ have been associated with genotoxicity at relatively low airborne concentrations (0.013–114 µg/m^3^) in production settings [1]. We employed higher concentrations in vitro to promote cellular uptake and accumulation in tissue niches; the observed stem-cell cytotoxicity supports further investigation of populations with elevated occupational or chronic cosmetic exposure [23].

Proliferation assays revealed concentration-dependent effects that partially align with the literature. For example, bone-marrow MSCs treated at 0.10 mg/mL showed reductions in proliferation of 10% (day 3), 40% (day 7), and 25.64% (day 14) [22], while HC3T3 murine osteoblasts and human enterocytes exposed to 10–50 µg/mL experienced 20–30% proliferation inhibition after 24 h [5]. In our system, treatment with 15.63 and 62.5 µg/mL produced a small, non-significant decline in proliferation (≈3–5%), whereas 250 µg/mL caused a 33% reduction after 7 days (Figure 3B). Differences among studies likely reflect nanoparticle dose, primary particle size, aggregation state, exposure time, and the assay methodology used.

Additionally, HaCaT cells were exposed to 25 µg/mL of TiO_2_ NPs (size: 18 nm) for 24 h [8], with neutral red staining revealing no adverse effects. Further assessment of proliferation using crystal violet staining after 7 days of treatment with 15.63 and 62.5 µg/mL of TiO_2_ NPs (size: 25 nm) showed a non-significant 5% decline. However, treatment with 250 µg/mL of these NPs over the same duration resulted in a significant 33% reduction in proliferation compared to control samples (Figure 3B). This reduction was slightly lower than the 40% decrease previously reported through flow cytometry on day 7 of exposure. This discrepancy may arise from the different concentrations of TiO_2_ NPs used 50 µg/mL versus 100 µg/mL and the distinct evaluation techniques employed. The lack of a significant change in proliferation for the 15.63 and 62.5 µg/mL concentrations of TiO_2_ NPs corresponds with the data obtained from the neutral red test [16,17]. Despite the presence of TiO_2_ NPs in the hMB-MSCs, in our system, cell viability and proliferation were little affected (Figure 3A,B), however, these TiO_2_ NPs are accumulated in the cytosol but when evaluated with transmission electron microscopy, the damage suffered by the hMB-MSCs by the presence of TiO_2_ NPs is evident, as severe damage to the cell membrane and organelles is observed (Figure 4E–H).

TiO_2_ NPs have been localized in various cell lines, revealing important insights into their cellular behavior. For example, in murine bone marrow osteoblasts, TiO_2_ NPs (20 μm) were detected within both the cytoplasm and the nucleus. In Wistar rat bone marrow mesenchymal stem cells (BM-MSCs), confocal microscopy indicated the presence of TiO_2_ NPs (108 and 196 nm) solely in the cytoplasm, with no contact observed with the nucleus [3]. Similarly, in MSCs derived from human bone marrow, when exposed to TiO_2_ NPs (50 nm), these particles were distributed throughout the cytoplasm but did not encroach upon the nuclear membrane [19]. Additionally, TiO_2_ NPs of 18 nm were internalized by HaCaT cells through active intracellular transport, being enclosed in double-membrane cytoplasmic compartments that resemble autophagosomes. NanoSIMS analysis corroborated that these NPs did not traverse the nuclear membrane [8].

Ultrastructural analysis demonstrated intracellular accumulation of TiO_2_ aggregates in hMB-MSCs. TEM images showed aggregates (~241 nm) localized in the cytosol and within vacuoles, but not penetrating the nuclear membrane, suggesting that they crossed the cell membrane, potentially via receptor-mediated endocytosis [5]. This pattern agrees with multiple reports showing cytoplasmic localization of TiO_2_ aggregates in diverse cell types [3,5,8,19]. Aggregate sizes vary with dispersion medium: proteins in FBS can limit agglomeration, while PBS favors larger clusters; this likely explains differences between TEM aggregate sizes and suspension measurements reported elsewhere [5,20]. Both in vivo and in vitro studies have established that the interaction of TiO_2_ NPs with cells induces the synthesis of pro-inflammatory cytokines in vivo and in vitro [18,24,25,26,27], as well as promoting the production of reactive oxygen intermediates (ROS), subsequently leading to oxidative stress [3].

Previous studies have reported an increase in ROS following the exposure of eukaryotic cells, thereby supporting the plausibility of apoptosis [28,29,30,31]. In contrast, our results demonstrated a significant decrease in ROS levels. Although this effect cannot be fully explained, the concentration employed allows us to speculate on a potential organelle dysfunction, particularly at the mitochondrial level [32], which could disrupt the biochemical mechanisms of the cell and compromise overall cellular homeostasis. Further investigations are warranted to clarify this observation.

Genotoxicity observed in the HepG2 cell line appears to be linked to the free radical production induced by TiO_2_ NPs. As DNA damage was associated with an increased generation of ROS, including H_2_O_2_ and hydroxyl radicals (•OH), alongside the release of cytochrome c and the activation of caspase-3 [33]. TiO_2_ NPs measuring 26.4 ± 1.2 nm have been shown to trigger ROS production in immune cells, including monocyte-macrophages and neutrophils [3]. Furthermore, activation of human neutrophils by TiO_2_ NPs suggests that, in some cases, the activation of the NADPH-oxidase enzyme complex contributes to ROS production, generating O_2_•, hydroxide ions (OH^−^), and H_2_O_2_ [34]. The current study raises pertinent questions regarding the observed reduction in ROS generation in response to TiO_2_ NPs (Figure 5). Future research will aim to elucidate this phenomenon, potentially utilizing inhibitors of the mitochondrial respiratory chain (e.g., sodium azide) or phagocytosis (e.g., cytochalasin-D) to gain insights into the mechanisms underlying the reduction in ROS production. The polymerization and depolymerization of actin filaments, as demonstrated in human and murine glial cells [2,35,36], may also play a significant role in this context.

Mesenchymal stem cells, like cells from any organ or tissue, express specific surface antigens on their membranes, notably CD73, CD90, and CD105. In human umbilical vein, endothelial cells (HUVEC), TiO_2_ NPs at concentrations ranging from 5 to 20 µg/cm^2^ have been documented to induce the expression of adhesion molecules such as E-selectin, P-selectin, intercellular adhesion molecule (ICAM), vascular cell adhesion molecule (VCAM), and platelet endothelial cell adhesion molecule-1 (PECAM-1) [37]. In the present study, TiO_2_ NPs with a size of 25 nm were shown to decrease the expression of CD73 (7.09%) and CD90 (25.73%) on the cell membranes of human MB-MSCs (Appendix A). CD90 is a critical membrane antigen that facilitates the adhesion of MSCs, leukocytes, and fibroblasts to endothelial cells. The expression of this antigen increases when endothelial cells are activated and interact with the leukocyte integrin Mac1 (CD11b/CD18), which plays a pivotal role in the homing and recruitment of immune cells [38]. Given the vital role of MSCs in cellular regeneration and tissue repair, alterations induced by TiO_2_ NPs may contribute to tissue injury and cellular dysfunction. Recent findings from our research group indicated that the treatment of umbilical cord-derived MSCs with TiO_2_ NPs at a concentration of 500 µg/mL resulted in a substantial decrease in cell viability, proliferation, and differentiation capacity toward adipocytes and osteoblasts, with reductions of approximately 60% and 20%, respectively. These results substantiate the findings presented in this study [38]. The cytotoxic effects of NPs can profoundly affect critical cellular outcomes, including the cell cycle, proliferation, differentiation, and programmed cell death. Specifically, TiO_2_ NPs can activate four distinct pathways of cell death: necrosis, apoptosis, necroptosis, or autophagy, depending on their physicochemical properties such as dosage, size, and shape [30]. In this paper, we demonstrate that TiO_2_ NPs do not induce the generation of ROS (Figure 5).

Our TEM data and functional readouts suggest that intracellular TiO_2_ accumulation can disrupt membrane integrity and organelles at higher concentrations, potentially triggering cell death mechanisms (necrosis, necroptosis, apoptosis, or autophagy) depending on nanoparticle physicochemical properties and dose [30]. In this study, we did not detect ROS-dependent signaling as a primary mechanism; therefore, cell death pathways independent of ROS—such as necrosis, necroptosis, or autophagy—are plausible and require targeted assays to confirm.

Our results showed that cellular viability and proliferation were only slightly altered; however, their effects led to a reduction in membrane markers. This suggests that molecules such as CD73, CD90, and CD105 may either undergo constant renewal or (2) be degraded at the membrane. On the other hand, if TiO_2_ NPs accumulate in the cytosol, they can interfere with multiple intracellular processes, including signaling pathways, ROS generation, cytokine synthesis, and other functional activities. In our system, the concentrations of TiO_2_ NPs used may have been too low to directly disrupt or inhibit processes such as ROS production. It is known that ROS generation involves organelles such as lysosomes and enzymes, including cathepsins, superoxide dismutase, as well as H^+^ ions stored within vacuoles. If these vacuoles also serve as reservoirs for TiO_2_ NPs, it is unlikely that they are activated or able to act directly on the NPs. Altogether, these mechanisms—or potentially other unknown ones—may be contributing to the cellular response observed upon TiO_2_ NP accumulation.

Autophagy induction can result in either cell survival or cell death, depending on the specific cell type and the stimuli involved. NPs have the potential to induce autophagic cell death by modulating the mTOR pathway, or they may obstruct autophagy through lysosomal impairment [28]. In our study, we highlight the toxic effects of TiO_2_ NPs on hMB-MSCs. These results have potential implications for regenerative medicine. MSCs reside in tissue niches and perform key roles in repair and immunomodulation. Disruption of MSC function by nanoparticle accumulation could impair tissue regeneration and compromise cell-therapy outcomes in environments with high nanoparticle exposure. However, translating in vitro concentrations to in vivo risk requires caution; further in vivo and mechanistic studies are essential to define real-world relevance.

## 4. Materials and Methods

### 4.1. Titanium Dioxide NPs

TiO_2_ NPs (Sigma-Aldrich, St. Louis, MO, USA) with a primary particle size of 25 nm, density 4.26 g/mL at 25 °C, pH 3.5–4.5, and primarily anatase phase (see Appendix A) were used. NPs were sterilized by autoclaving at 1.5 atm for 20 min. A sterile stock suspension of TiO_2_ NPs (1 mg/mL) was prepared in PBS (150 mM NaCl, 4.4 mM KCl; pH 7.4) and stored at 4 °C. Before each experiment, aliquots of the TiO_2_ stock were centrifuged at 750× *g*, the supernatant was removed, and the pellet was resuspended in DMEM/F12 supplemented with 10% FBS to obtain the working suspensions that were then added to cell cultures.

### 4.2. Characterization of TiO_2_ NPs

For imaging, TiO_2_ samples were prepared for scanning electron microscopy (SEM) after sputter-coating with gold. For transmission electron microscopy (TEM), TiO_2_ suspensions (1 mg/mL) were prepared in either PBS or DMEM/F12 + 10% FBS (Gibco, Waltham, MA, USA) and examined on a JEOL 10/10 transmission electron microscope. Phase composition was determined by X-ray diffraction (see Appendix A).

### 4.3. Human Menstrual Blood Mesenchymal Stem Cells

hMB-MSCs were isolated at the GemLab laboratory following informed consent and with approval from the Ethics Committee of the Escuela Superior de Medicina, Instituto Politécnico Nacional (approval code: ESM.CEI-02/31-05-2017) (see Appendix A). Healthy female donors (18–25 years), negative for HBV, HCV, syphilis, HIV, and HPV, provided menstrual blood samples collected during the first three days of menses. Samples were processed under sterile conditions in a laminar-flow hood. Menstrual blood was diluted with PBS, and micro-explants were seeded onto T-75 flasks for 15 min to allow initial adherence. Non-adherent material was removed and fresh DMEM/F12 containing 10% FBS and 1X antibiotic–antimycotic (Gibco) was added. Cells were cultured in DMEM/F12 plus 10% FBS at 37 °C, 5% CO_2_, and ≈90% relative humidity until ~80% confluence. Experiments were performed with cultures at passages 5–8, each experiment run independently in triplicate.

### 4.4. Differentiation Assays (Osteogenic and Adipogenic)

For differentiation, 2 × 10^3^ cells per well were plated in 24-well plates. Osteogenic differentiation was induced using StemPro osteogenesis basal medium supplemented with StemPro osteogenesis supplement (Gibco, ThermoFisher, Waltham, MA, USA) for 14 days. Adipogenic differentiation was induced using StemPro adipogenesis basal medium with the corresponding adipogenesis supplement (Gibco) for 14 days. After induction, cells were fixed with 10% neutral buffered formalin and processed for specific staining.

### 4.5. Staining for Osteogenic and Adipogenic Differentiation

Osteogenesis was evaluated by Alizarin Red S staining (Sigma, Aldrich, St. Louis, MO, USA). Following 14 days in osteogenesis medium, cells were washed with PBS and stained with 2% Alizarin Red S (pH 5.5) for 5 min to detect extracellular calcium deposits; excess dye was removed by washing, and samples were imaged under an optical microscope (Leica, incorporation, Wetzlar, Germany) at 10× (DP24 U/S digital camera). Adipogenesis was assessed by Oil Red O staining (Sigma, Aldrich, St. Louis, MO, USA). After 14 days in adipogenesis medium, cells were stained with Oil Red O for 10 min at room temperature, washed with 60% isopropanol, and imaged at 10×.

### 4.6. Cell Viability Assay

Cell viability was measured using the Alamar Blue assay [39]. Briefly, 5 × 10^3^ hMB-MSCs/well were seeded in 96-well plates. A 100 μL aliquot of TiO_2_ stock (1 mg/mL) was added to the first well (all wells contained 100 μL medium), and twelve serial 1:2 dilutions were performed across the plate to achieve a final range down to 0.0002 mg/mL; final well volumes were maintained at 100 μL. Each concentration was assayed in triplicate. Controls included: negative control (DMEM/F12 + 10% FBS) and positive control (DMEM/F12 without FBS + DMSO to induce cell death). Cells were incubated with TiO_2_ working suspensions for up to 14 days; fresh medium (50 μL) was added to each well every 3 days. After the incubation period, 50 μL of Alamar Blue diluted 1:9 (*v*/*v*) (Invitrogen, ThermoFisher, Waltham, MA, USA) in DMEM without phenol red was added to each well, and plates were incubated for 2 h at 37 °C. Absorbance was measured at 630 nm (Bio-Rad spectrophotometer, Hercules, CA, USA).

### 4.7. Cell Proliferation: Crystal Violet Assay

Proliferation was assessed by crystal violet staining. hMB-MSCs (5 × 10^3^ cells/well) were seeded in 96-well plates and allowed to attach for 8 h before treatment with TiO_2_ at 15.63, 62.5, or 250 μg/mL (or vehicle control) for 7 days. Fresh medium (50 μL) was added on days 3 and 6. Cells were fixed with 50 μL of 10% glutaraldehyde for 5 min, washed, air-dried, stained with crystal violet for 10 min (Sigma, Aldrich, St. Louis, MO, USA), washed, and the dye was solubilized with 50 μL of 10% acetic acid. Absorbance was read at 630 nm (Bio-Rad, Hercules, CA, USA) [40].

### 4.8. Expression of Membrane Markers

hMB-MSCs were seeded at 4 × 10^5^ cells/dish and exposed, or not, to 62.5 μg/mL TiO_2_ NPs for 24 h. Cells were detached with 2 mM EDTA in PBS (1X), washed twice with PBS, and incubated for 20 min at room temperature with FITC- or PE-conjugated monoclonal anti-human antibodies against CD73, CD90, CD105, CD14, CD34, and CD45 (Becton Dickinson, Franklin Lakes, NJ, USA). After two washes in cold PBS + 0.1% BSA, cells were fixed with 1% paraformaldehyde in PBS. A total of 1 × 10^4^ events were acquired from a predefined gate on a FACSAria II flow cytometer (Becton Dickinson, Franklin Lakes, NJ, USA) and analyzed with Flowing Software 7.6.2.

### 4.9. Semi-Quantitative Assay Using NBT Assay for Superoxide (ROS Proxy)

For the NBT assay, 2 × 10^4^ hMB-MSCs/well were seeded in 24-well plates and treated with 15.63, 62.5, or 250 μg/mL TiO_2_ NPs for 4 days. Cells were washed three times with sterile PBS and incubated with DMEM supplemented with 100 μL NBT (1.6 mg/mL) (Sigma, Aldrich, St. Louis, MO, USA) for 90 min at 37 °C. After incubation, wells were washed three times with PBS and formazan was solubilized by adding 100 μL of 2 M KOH and 100 μL DMSO per well. Absorbance was read at 630 nm (Bio-Rad, Hercules, CA, USA).

### 4.10. Statistics Analysis

Data are presented as mean ± SD. Statistical analyses were performed using one-way ANOVA in SigmaPlot v14.0. For viability and proliferation, statistical significance was set at *p* ≤ 0.001; for other assays, *p* ≤ 0.05. Exact tests, *n* (biological replicates), and post hoc comparisons are reported in the figure legends.

## 5. Conclusions

This is the first study to evaluate the effects of 25 nm TiO_2_ NPs (predominantly anatase) on hMB-MSCs. Following in vitro exposure, TiO_2_ NPs were internalized and accumulated in perinuclear regions and cytoplasmic vacuoles at the highest tested concentration (250 µg/mL). Exposure produced modest decreases in cell viability and proliferation and was accompanied by reduced membrane expression of CD73 and CD90, together with an unexpected reduction in ROS production. Collectively, these findings indicate that TiO_2_ exposure can alter key hMB-MSC functions, including immunomodulatory behavior, differentiation potential, and regenerative capacity, and warrant further mechanistic and in vivo investigation.

## Figures and Tables

**Figure 1 ijms-26-11168-f001:**
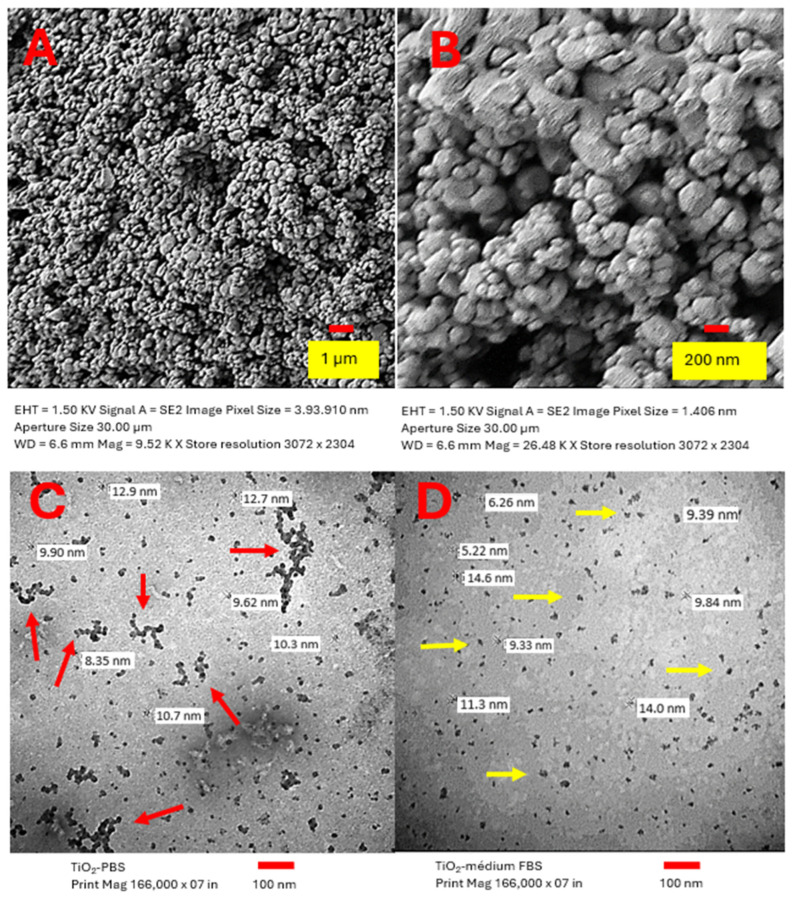
Characterization of TiO_2_ NPs: (**A**,**B**) Scanning electron microscopy (SEM) images of powdered TiO_2_ NPs showing the primary-particle morphology. (**C**) Transmission electron microscopy (TEM) image of TiO_2_ NPs suspended at 1 mg/mL in phosphate-buffered saline (PBS). (**D**) TEM image of TiO_2_ NPs suspended at 1 mg/mL in DMEM/F12 supplemented with 10% fetal bovine serum (FBS). Note the reduced agglomeration in the serum-containing suspension. (SEM/TEM imaging performed on JEOL instruments; see Section 4 for preparation and imaging parameters).

**Figure 2 ijms-26-11168-f002:**
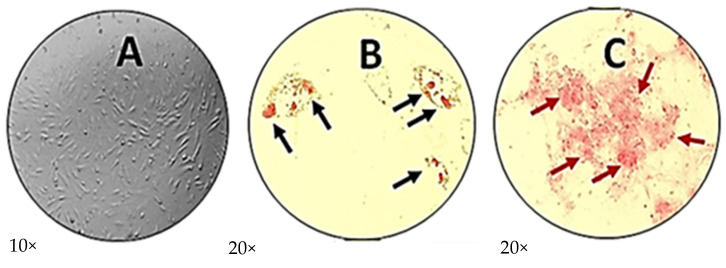
Characterization in vitro of human menstrual blood-derived mesenchymal stem cells (hMB-MSCs): (**A**) Phase-contrast image of cultured hMB-MSCs showing spindle-shaped, flat, fibroblast-like morphology (10×). (**B**) Adipogenic differentiation: Oil Red O staining reveals intracellular lipid droplets (black arrows) after 14 days in adipogenic medium (20×). (**C**) Osteogenic differentiation: Alizarin Red S staining identifies extracellular calcium deposits (red arrows) after 14 days in osteogenic medium (20×). Cells were differentiated for 14 days; representative fields are shown.

**Figure 3 ijms-26-11168-f003:**
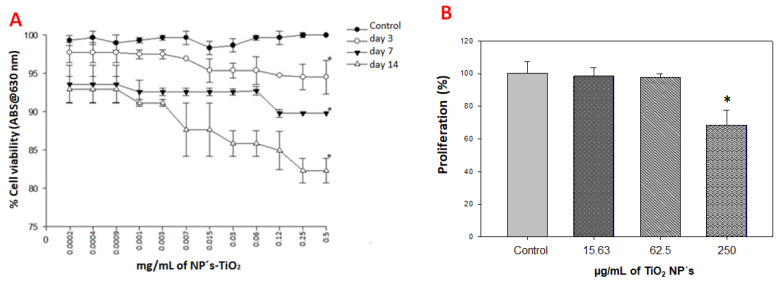
TiO_2_ NPs reduce hMB-MSC viability and proliferation in a dose-dependent manner: (**A**) Cell viability measured by Alamar Blue at 3, 7, and 14 days following exposure to increasing concentrations of TiO_2_ NPs. High concentrations (250 µg/mL) produced the largest reduction in viability at day 14. (**B**) Cell proliferation measured by crystal violet staining after 7 days of exposure to 15.63, 62.5, and 250 µg/mL TiO_2_ NPs. Data are presented as mean ± SD (*n* = 6). * *p* ≤ 0.001 versus control (one-way ANOVA).

**Figure 4 ijms-26-11168-f004:**
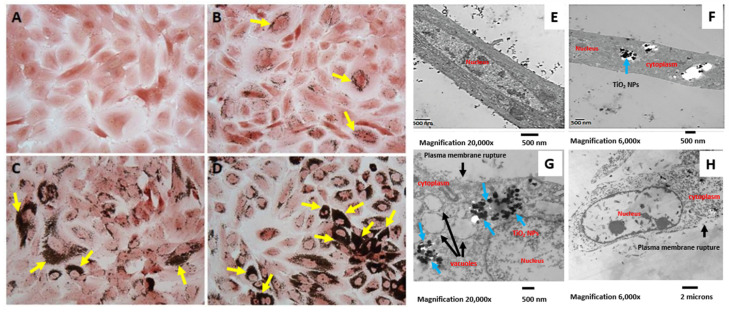
Internalization and intracellular localization of TiO_2_ NPs in hMB-MSCs: (**A**–**D**) Light-microscopy images (safranin stain) of hMB-MSCs after 7 days in DMEM/F12 (**A**) or DMEM/F12 containing 15.63 µg/mL (**B**), 62.5 µg/mL (**C**), or 250 µg/mL (**D**) TiO_2_ NPs. Yellow arrows indicate cytoplasmic nanoparticle aggregates adjacent to the nucleus (20×). (**E**–**H**) Electron microscopy images of cells cultured for 3 days showing intracellular aggregates and vacuolization. Aggregates (blue arrows) are observed within cytoplasmic vacuoles (black arrows); at the highest dose (250 µg/mL), membrane and organelle disintegration is evident.

**Figure 5 ijms-26-11168-f005:**
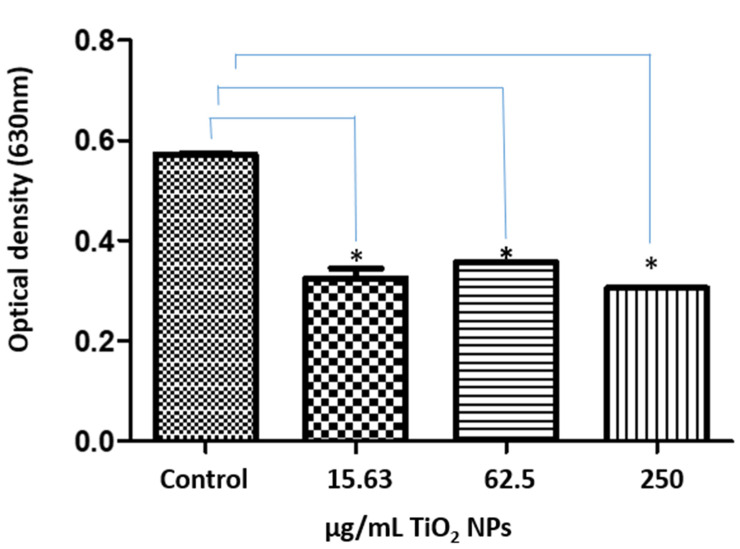
Reactive oxygen species (ROS) production in hMB-MSCs exposed to TiO_2_ NPs. NBT reduction (proxy for superoxide/NADPH-oxidase activity) measured after exposure to 15.63, 62.5, and 250 µg/mL TiO_2_ NPs. Treated cells exhibited a marked decrease in NBT reduction relative to control (*n* = 3). Error bars indicate SD. * Statistical difference vs. control determined by one-way ANOVA (*p* < 0.005).

## Data Availability

Data are available upon request from the corresponding author.

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
