# Peer review of "Toxicological Effects of Titanium Dioxide Nanoparticles on Human Menstrual Blood Mesenchymal Stem Cells"

_ijms, 2025, doi:10.3390/ijms262211168_

Round 1
Reviewer 1 Report
Comments and Suggestions for Authors
The manuscript, titled "TOXICOLOGICAL EFFECTS OF TITANIUM DIOXIDE NANOPARTICLES ON HUMAN MENSTRUAL BLOOD MESENCHYMAL STEM CELLS” addresses a very important current topic, the potential toxic effects of titanium oxide nanoparticles on human health, focusing its study on human menstrual blood mesenchymal stem cells. Some minor corrections and additional major clarifications are required before final approval.
Minor issues:
1-Lines 30-31: Improve the sentence, something like this: ..nanometric size (<100 nm) allows it to enter the bloodstream, from where ......
2-Line 34: TiO2, throughout the manuscript, the number should be in subscript (TiO2).
3-Line 41: add…..15.63, 62.5 and 250 µg/mL “of TiO2”.
4-Line 42: remove TiO2 from the sentence.
5-Line 44: delete the second word “marker”.
6-Line 45: TiO2 NPs without, change for “without TiO2 NPs”.
7-Lines 41 and 46: define exactly what concentration is used, 15.62 or 15.63, the same throughout the entire manuscript.
8-Figure 2: The figure says "flujo menstrual" in Spanish.
9-Line 154: Throughout the manuscript there is the word nanoparticles and the abbreviation NPs.....just write the word the first time with the abbreviation in parentheses and then only use the abbreviation.
10-Line 164: missing parenthesis.
11-Line 173: the same as line 154 for dioxide nanoparticles (TiO2 NPs).
12-Line 192: What does "**" mean?.
13-Figure 4: talks about red arrows, but in figure no red arrow appears.
14-Figure 5: The figure says "flujo menstrual" in Spanish.
15-Figure 6 has the x-axis references shifted
16-Line 273: um or nm?
17-Line 277: Miscited Bo et al.
18-Line 289: in vivo and in vitro must be in italic across the manuscript.
19-Line 295: I think it's better to change the wording to "aggregates of approximately 241 nm in size were formed".
20-Line 375: Titanium dioxide nanoparticles are repeated.
21-Line 376: delete "be"
22-Line 378: minutes always write the same...minute or min
23-Line 381: “NPs were suspended” instead “suspended”.
24-Line 402 is the cell concentration per well?
25-Line 415: replace "will be" with "was".
26-Lines 439-440: “in the presence or not of……” The same for line 450.
27-Line 449: Clarify what NBT means.
28-Line 459: “with ONE WAY…”
Major issue:
29-In 2.5. section, “Expression of membrane markers on hMB-MSCs treated with TiO2 NPs” Why did they authors choose only that concentration (62.5)?
30- In Figure 5, the panels are inverted. The text states that treatment with nanoparticles decreases two markers (CD90 and CD73), but according to the figure, this would be an inverse result. How is this actually the case?
31-Line 248: Why do the authors claim differences according to the size of the nanoparticles when what they are doing is comparing between concentrations?
32-General: When making comparisons with other authors it is important to compare both the system used and the concentration and size of the nanoparticles.
33-Lines 263-265: I don't understand the sentence.
34-Line 320: I'm not sure that ROS inhibition can be claimed, that would have to be tested.
35- Line 329: NPs are claimed to be 25 nm, but the results indicate an average of 19 nm. DLS results should be incorporated as figure.
36-Lines 386-387: I don't understand the context of this sentence.
37-Line 393: Eliminate Culture of…..
38-Line 442: What do saturated concentrations mean? This can lead to false positives...
39-General: How cell proliferation was measured and how to differentiate it from cell viability are not adequately explained, both in the methodology and in the description of the results and the corresponding figure.
40-The authors should discuss the relationship between alterations in viability and proliferation. And why they believe that there is a possible reduction in the activation of mechanisms that would lead to the generation of free radicals.
41-Only 4 works from the last 5 years are cited, out of a total of 29. More recent citations should be included, at least another 5.
Comments on the Quality of English LanguageMany of the minor issues have to do with writing.
Reviewer 2 Report
Comments and Suggestions for Authors
A human biomaterial was used in the article, so it is necessary to provide the patient's consent.
Line 46 what did the authors mean by the abbreviation ROI?
The Conclusion section. In the abstract, it needs to be expanded to more than 1 sentence.
More detailed information about the characteristics of nanoparticles is required. What is the size according to the TEM data and what is the original zeta potential before the suspension is formed in a culture medium with serum?
Figure 2. It seems to me that the signatures of specific markers on flow cytometry rafts should be increased for better perception.
Cytotoxicity and proliferation analysis was performed over a fairly long period (several days) of incubation. Specify in the materials and methods whether the culture medium was replaced during co-incubation with nanoparticles?
Figure 6. The authors write about the evaluation of free radicals. Which ones exactly?
4.8 Semi-quantitative assay using NBT. The authors used this method to evaluate the generation of ROS, which looks strange, since this dye is not usually used to analyze the level of intracellular ROS. I suggest conducting a classic test using DCF, DHE, Cell ROX or other dyes widely used for these purposes.
In the conclusions section, it is necessary to provide a broader overview of the results obtained and briefly disclose the possible mechanisms of the identified cytotoxic effects of nanoparticles.
Figure 5. It requires reworking because the presented graphs are poorly readable and difficult to perceive due to the small font.
Figure 4. There is no scale bar in the microphotographs, it must be added. The resolution of the images is very low, so we need to update them.
Figure 3. How was the rate of cell proliferation assessed? There is no information about the assessment methodology.
Line 193 it is necessary to adjust the caption to Figure B
The quality of all drawings needs to be improved. This resolution of the drawings is unacceptable.
It is also necessary to rewrite the conclusions and expand them in more detail.
Reviewer 3 Report
Comments and Suggestions for Authors
This manuscript investigates the cytotoxic and functional effects of titanium dioxide nanoparticles (TiO₂ NPs) on human menstrual blood-derived mesenchymal stem cells (hMB-MSCs). The study focuses on cell viability, proliferation, marker expression, nanoparticle internalization, and reactive oxygen species (ROS) generation. This well-conceived, methodologically sound study fills a notable gap in nanotoxicology research by examining a clinically relevant, rarely studied stem cell population.
The data convincingly demonstrate the concentration- and time-dependent toxicity of TiO₂ NPs on hMB-MSCs, including altered marker expression and suppressed ROS production. This study provides valuable insights into how exposure to nanoparticles may impact stem cell behavior, especially in regenerative medicine contexts where hMB-MSCs are being explored for therapeutic use.
The highest concentration used (250–500 µg/mL) may exceed physiologically relevant exposure levels in most real-world scenarios. The study would benefit from a discussion of how these doses relate to occupational or clinical exposure levels. Additionally, pharmacokinetic modeling or citation of in vivo exposure data would justify these in vitro concentrations.
One key and unexpected finding is the decrease in ROS production with TiO₂ NP exposure. This finding contrasts with prior literature, which more often shows increased oxidative stress. Including a brief section that discusses possible mechanisms, such as mitochondrial suppression, enzyme inhibition, and endocytosis sequestration, would enhance the manuscript. Even if speculative, it would provide context for the results.
The study reports viability loss and organelle disruption, yet it fails to distinguish between apoptosis, necrosis, and autophagy. While this may be outside the scope of the current study, the discussion should acknowledge this limitation and suggest investigating it in the future.
Flow cytometry results show decreased CD73 and CD90 expression after exposure. While statistically sound, it's unclear whether this reflects a phenotypic shift, cellular stress, or loss of cell identity. It would be valuable to include a short paragraph addressing what this reduction implies in terms of MSC functionality.
Please clarify whether a positive control for ROS (e.g., H₂O₂) or marker expression was included.
The origin of the hMB-MSCs is unclear. Please briefly describe the isolation procedure, the number of donors, and their characteristics, as well as any other relevant information.
The panels in figures 2D, 5A, and 5B are too small and lack sufficient resolution. I suggest moving these figures to the supplementary materials and using bar plots showing the percentage of expression of each membrane marker instead.
Figure 3A should be redone. The Y-axis has an error in labeling and scaling, showing only 0%, 60%, and 100%. Similarly, the X-axis scaling could be appropriate for a bar plot but not for scatter and line plots because all tested doses are represented with the same distance between them, even though they are not in an arithmetic progression (e.g., the distance between 0.2 and 0.4 µg/mL is the same as the distance between 0.4 and 0.9 µg/mL, which is the same as the distance between 0.9 and 1.9 µg/mL, etc.).
In Figure 3B, the 250 µg/mL label is not centered below the corresponding tick. The same problem occurs in Figure 6 with the labels for 15.63, 62.5, and 250 µg/mL.
The pictograms in Figure 4 are too small. These photographs should be at least three times larger.
In Figures 2, 5, and 6, please state the number of independent determinations (n).
Round 2
Reviewer 1 Report
Comments and Suggestions for Authors
The manuscript has been greatly improved by the authors, but several more corrections still need to be made before it is accepted for publication. It was very difficult to analyze the changes in the manuscript as they were submitted. It is important to indicate the changes made and the new line numbers where they are located, given the shift in the text.
- In the supplementary figures related to flow cytometry (previously Figures 2 and 5 in the manuscript), Spanish words are still present. I understand that each panel originates in this way by the software, but I believe the authors should try to eliminate them, if possible.
- Throughout the manuscript, the word "nanoparticles" continues to appear several times instead of the abbreviation NPs.
- Figure 5 was removed and incorporated as a supplementary figure, but the manuscript contains Figures 4 and 6, but not Figure 5. Please revise.
4- Line 415: “The vast majority of the global population is way exposed to contact with TiO2 NPs in one of their forms”. Delete the word "way."
5- Line 521: As in the previous revision of the old line 273, the µm for nm ratio was not corrected.
6- Lines 506 and 508: Correct TiO2 for TiO2.
7- In Materials and Methods, there are two items as 4.6.
8- As noted in the previous revision, in Figure 5, the panels are inverted. The text indicates that nanoparticle treatment decreases two markers (CD90 and CD73), but according to the figure, this would be the opposite result. Figure 5 is now a supplementary figure, but I noticed that it has not changed from the original to align with the manuscript text. On the other hand, I repeat, Figure 5 does not appear in the manuscript.
Author Response
Mexico City, October 03, 2025
Dear Editor
The authors of the manuscript entitled “Toxicological Effects of Titanium Dioxide Nanoparticles on Human Menstrual Blood Mesenchymal Stem Cells” (ID: ijms-3804280), submitted to the International Journal of Molecular Science, sincerely appreciate the reviewers’ valuable comments and suggestions, all were performed and are labeled with a yellow etiquet. These insights have significantly contributed to enhancing the quality of our work. We have carefully addressed all concerns raised, and the revisions have been implemented as detailed below.
Thanks for all observation.
Sincerely
Gisela Gutierrez-Iglesias
Corresponding.
Reviewer 1
- In the supplementary figures related to flow cytometry (previously Figures 2 and 5 in the manuscript), Spanish words are still present. I understand that each panel originates in this way by the software, but I believe the authors should try to eliminate them, if possible.
We appreciate your observation. The Spanish words previously present in the flow cytometry panels have been removed. - Throughout the manuscript, the word "nanoparticles" continues to appear several times instead of the abbreviation NPs.
Thank you for pointing this out. The term "nanoparticles" has been consistently replaced with the abbreviation "NPs" throughout the manuscript. - Figure 5 was removed and incorporated as a supplementary figure, but the manuscript contains Figures 4 and 6, but not Figure 5. Please revise.
Thank you for your observation.
The figures have been renumbered to ensure continuous and logical numbering throughout the manuscript.
- Line 415: “The vast majority of the global population is way exposed to contact with TiO₂ NPs in one of their forms”. Delete the word "way."
We are grateful for this correction. The manuscript was reviewed by a native English speaker (certificate included), and typographical errors such as this have been corrected accordingly. - Line 521: As in the previous revision of the old line 273, the µm for nm ratio was not corrected.
We appreciate your careful review. All measurements related to particle size have been corrected to consistently use nanometers (nm). - Lines 506 and 508: Correct TiO2 for TiO₂.
Thank you for this suggestion. The chemical formula has been correctly formatted throughout the text using the appropriate subscript notation (TiO₂).
- In Materials and Methods, there are two items as 4.6.
Thank you for noticing this error. The subsection numbering in the Materials and Methods section has been revised to avoid duplication.
- As noted in the previous revision, in Figure 5, the panels are inverted. The text indicates that nanoparticle treatment decreases two markers (CD90 and CD73), but according to the figure, this would be the opposite result. Figure 5 is now a supplementary figure, but I noticed that it has not changed from the original to align with the manuscript text. On the other hand, I repeat, Figure 5 does not appear in the manuscript.
Thank you once again for your detailed comment. The supplementary figure (previously Figure 5) has been updated to accurately reflect the data described in the manuscript. In addition, the figure has been properly referenced and labeled to ensure consistency throughout the document.
Reviewer 3 Report
Comments and Suggestions for Authors
I have carefully evaluated the revised version of the manuscript along with the authors’ rebuttal letter. The authors have made substantial improvements in response to my previous comments.
The discussion has been expanded to include the physiological relevance of the tested TiO₂ concentrations, the unexpected reduction in ROS, and the functional implications of reduced CD73/CD90 expression.
The limitations regarding apoptosis, necrosis, and autophagy have been acknowledged, and future directions are outlined.
Overall clarity, transparency, and scientific rigor have been significantly enhanced.
The only remaining limitation is the absence of a positive control in the ROS assay. The authors acknowledge this limitation and justify their decision by referencing prior literature. While this remains a methodological weakness, I do not believe it precludes publication.
Conclusion: The authors have successfully addressed my concerns. I recommend the manuscript for publication in its present form.
Author Response
Reviewer 3
I have carefully evaluated the revised version of the manuscript along with the authors’ rebuttal letter. The authors have made substantial improvements in response to my previous comments.
We sincerely thank you for your thorough review and positive evaluation of our revised manuscript.
The discussion has been expanded to include the physiological relevance of the tested TiO₂ concentrations, the unexpected reduction in ROS, and the functional implications of reduced CD73/CD90 expression.
We appreciate your recognition of these additions. We believe they enhance the depth and clarity of our discussion.
The limitations regarding apoptosis, necrosis, and autophagy have been acknowledged, and future directions are outlined.
Thank you for noting this. We aimed to provide a transparent and critical assessment of our study’s scope and limitations.
Overall clarity, transparency, and scientific rigor have been significantly enhanced.
We are grateful for your encouraging feedback.
The only remaining limitation is the absence of a positive control in the ROS assay. The authors acknowledge this limitation and justify their decision by referencing prior literature. While this remains a methodological weakness, I do not believe it precludes publication.
We thank you for your understanding regarding this limitation. We agree that while the absence of a positive control is a weakness, it does not undermine the overall validity of our findings.
Conclusion: The authors have successfully addressed my concerns. I recommend the manuscript for publication in its present form.
We deeply appreciate your recommendation and the time you dedicated to reviewing our work. Your insights have been instrumental in improving the quality of the manuscript.